# Food Waste in Distribution: Causes and Gaps to Be Filled

**Francisco Carlos Vaz Sales, Michele De Souza, Luiz Reni Trento** [ID]**, Giancarlo Medeiros Pereira** *[ID]**,
Miriam Borchardt and Gabriel Sperandio Milan** [ID]

Programa de Pós Graduação em Engenharia Industrial, Escola Politécnica, Campus São Leopoldo,
Universidade do Vale do Rio dos Sinos-UNISINOS, S. Leopoldo 93022-750, RS, Brazil
* Correspondence: gian@unisinos.br

**Abstract:** This qualitative study investigated the gaps that hinder fruit and vegetable waste reduction in small distributors serving the last miles of the food chain. Fifteen Brazilian distributors operating far from the producers were analyzed. The findings contribute to the literature by showing several research gaps. The surplus in farmer planting increases waste generation at the level of distributors. We should know how to collect and process the relevant data to forecast the demand of each small farmer or distributor (e.g., tendencies in market demands or other farmers' planting plans). Sectoral entities should use these data to help actors define how much to plant or buy. The acceptance of waste by farmers and distributors has a financial reason. Changing such acceptance requires the demonstration of financial gain that a more sustainable approach may have. We need to know how to calculate the economic gains and losses related to waste reduction throughout the chain (before developing useful mitigators). We should also know how to induce entrepreneurs to invest in better resources or practices in transportation, handling, packaging, and storage. Selling items before their decline avoids waste. We need to know how to improve small actors' gains to increase sales of such products.

**Keywords:** food; supply chain; sustainability; waste; retail

## 1. Introduction

Brazil is one of the ten largest producers of food waste globally. Approximately 30% of all food produced (about 40,000 tons) goes to waste [1]. Due to their low perishability, fruits and vegetables (F&V) represent the most discarded items, accounting for up to 66% of food waste [2,3]. Waste is estimated at 9.5 tons per week and is predominantly composed of bananas, papayas, tomatoes, peppers, and lettuce [4]. This accounts for a third of global production, posing severe threats to food security and the long-term economic sustainability of the supply chain [4,5]. Waste can occur from producer farms to consumers, passing through the entire FSC (food supply chain) [6].

Waste is an unacceptable loss, because the production of a discarded food uses scarce natural resources (e.g., water, fertile soil, energy, and various inputs) [7,8]. Such waste becomes even more unacceptable when it is known that approximately 931 million tons of food were thrown away in 2019, while 690 million people are hungry. It is estimated that up to 3 billion could be deprived of food in the post-COVID-19 period [2]. So, social losses are relevant, as wasted food could have been used to improve the nutrition of people with lower purchasing power [9].

Food insecurity may increase in the coming years. Estimates indicate that the global population will increase from 7.7 billion today to 9.7 billion in 2050 [10]. Furthermore, the growth is estimated to be as high as 2 billion people in less than 30 years (+/−25% of today's population). This implies a considerable growth in the demand for food and represents a significant challenge for the agricultural sector [11], especially when considering that the pressure for more food is not accompanied by a proportional expansion of the cultivable area [12,13]. In addition, F&V are sensitive products that can suffer damage

due to mishandling, leading to considerable losses [5,14]. Commercialization and the seasonality of prices also represent aggravating difficulties in accessing them [15].

Waste in the final miles of a food chain is a significant problem. Better management of transactions between farmers, and fruit and F&V distributors can mitigate part of this waste [1]. A lower level of losses could help improve product quality, mitigate environmental damage [6,16], and reduce food prices [15]. However, some gaps still need to be filled (before mitigation actions are defined). These gaps indicate that we need to better understand what causes waste in F&V distribution [1] and how to reduce such waste [17]. Thus, the following research questions are proposed:

RQ1: What causes waste in fruit and vegetable distribution?

RQ2: What should we know to reduce waste in F&V distribution?

Adverse impacts on environmental sustainability require FSC links, researchers, and sectoral entities to deal with the problem at all chain levels [15,17]. Investigating the causes and mitigators of produce waste to be readily applied by distributors could reduce the waste of these items still fit for human consumption, thus minimizing the unnecessary depletion of scarce natural resources [7]. Furthermore, it could also promote food security, promote sustainable agriculture, and improve nutrition [18].

The research question is addressed with a qualitative study on fifteen regional distributors of F&V in the city of Imperatriz in the State of Maranhão/Brazil. The main items sold are watermelons, bananas, tomatoes, onions, cabbage, papayas, and onions. The analysis of the data collected in the interviews applied to distributors reveals the causes of wastage of F&V and enables us to define mitigating measures for these causes. As well as the diagnosed causes, the manuscript indicates a number of mitigating proposals for them. These causes and mitigators were organized in the categories of farmer production exceeding the demand, transportation/handling, storage/packaging, and sales below expectations.

## 2. Methods

The manuscript presents a literature review focusing on the F&V production chain and food waste. This literature review enabled us to structure an investigation to contribute to the research objectives.

### 2.1. Literature Review

2.1.1. Food Waste Scenario

While 690 million people go hungry every day, approximately 931 million tons of food were wasted on the planet in 2019, 61% at the domestic level, 26% in food services, and 13% in retail [2]. Part of this waste occurs in the initial stage of the production chain [19]. Household consumption generates 50% of FW, which contributes to an inevitable mismatch between the decision to avoid waste for monetary, environmental, and moral reasons, and what the consumer does (despite awareness of the urgent need to reduce the undesirable effects of human behavior on ecosystems) [8].

Therefore, countries need to ensure food security for the growing human population. However, such provision must be ensured without widespread environmental degradation, since the agrifood system contributes significantly to greenhouse-gas emissions. In addition, we must consider the most significant sources of pollution of ecosystems, waterways, and oceans. Among these sources, stakeholders in agricultural production need to pay attention to the vast amounts of pesticides applied in agricultural production or the depletion of non-renewable resources [7,17].

2.1.2. Food Supply Chain

The food supply chain (FSC) is defined as the sequence of processes centered on materials, information, and cash flow. The FSC comprises the production, transportation, distribution, retail, consumption, and disposal of products through the employment of interdependent human and natural resources [20].

The F&V supply chain covers production and distribution through the execution of activities distributed in several stages (e.g., farms, distributors, wholesalers, retailers, and the consumer's table) [21]. In this supply chain, many product-related challenges exist (e.g., shelf-life constraints, quality and quantity variability, and production yield). Other issues include biological variations, seasonality, and random factors related to weather, pests, and biohazards [22,23]. Located between producers and retailers, the distribution sector represents an under-investigated stage of the F&V chain. As a result, few reliable data or practices regarding distributors are available [5].

### 2.1.3. Causes of Waste

In this supply chain, many product-related challenges exist (e.g., shelf-life constraints, quality and quantity variability, and production yield). Other issues include biological variations, seasonality, and random factors related to weather, pests, and biohazards [24]. Details are provided below.

F&V waste seems to start in farms. Planting inappropriate varieties, incorrect sowing dates, and the lack of consideration of environmental factors can produce items of poor quality. Low-quality agricultural products can turn into waste rapidly [25]. Other authors highlight that farmer practices or options can also leverage waste. The literature cites improper harvesting techniques, poor handling inside the farm, the selection of less qualified transporters (who can cause damages during transportation to the distributors), and insufficient packaging [22,24,26].

The causes of F&V waste in distribution are related to the mishandling or lack of care in the cold chain [6,27,28]. The literature also presents other problems at the distributor level: inadequate packaging or storage facilities [22,23,27], lack of coordination and information sharing in the supply chain [29], inadequate transportation to retail facilities [30], and inefficient warehouse management [31–33].

In the last miles of the F&V chain, infrastructure problems occur in 75% of sales locations [4]. At the point of sale, fresh products can be damaged when displayed for sale [23,31], or during handling or storage [4,22]. Besides these problems, losses also occur due to inadequate storage and a lack of attention paid to conservation and refrigeration [4,31]. As for consumers, losses can occur due to excessive purchases and poor storage conditions [34,35], lack of care while handling [4], and preference for fresher, more perfect items [32].

F&V losses can be linked to deficiencies in awareness and training throughout the chain. These deficiencies can generate mechanical damage or promote microbiological problems that speed up the deterioration of F&V in the warehouse or on the shelf [4].

### 2.1.4. Waste Mitigation

Mitigating waste in the distribution stage requires training employees on the best handling practices. Such training seems to be a mandatory first action [24]. Distributors need to pay attention to the stages of the products during storage in the warehouse. One study identified that F&V waste could be significantly reduced in the pre-storage phase [36]. The improvement of operational management should focus on controlling food surpluses [37] and on the optimization of infrastructure, management, and conservation practices [4,15,28,38]. Distributors should also improve their decision making regarding the quantities of items purchased or their influence on turnover [5]. Additionally, distributors should cooperate and exchange information with all stakeholders in the F&V chain [39,40].

At the end of the F&V chain, the literature indicates the importance of packaging for the durability of fresh product guarantee [4]. Product durability could be increased by developing intelligent packaging, with resources for monitoring safety and quality, and techniques for preserving the useful life of these products. Adjusting the packaging dimensions to the needs of consumers [41,42] or paying attention to the accuracy in recording the expiration date is also suggested [42]. Furthermore, organizing joint promotions

or improving forecasts and orders in the chain could also mitigate waste throughout the chain [23,43,44].

### 2.2. Summary of the Literature

Food waste occurs in all chain stages (planting, transporting, handling, and marketing). According to the literature, mitigating waste in the distribution stage requires training employees on the best handling practices or controlling food surpluses. Distributors should also improve their decision making regarding the quantities of items purchased. Furthermore, organizing joint promotions or improving forecasts and orders in the chain can also mitigate waste.

### 2.3. Research Design

This study adopted a qualitative approach to identify the causes and gaps that prevent mitigating such reasons in the last miles of the chain. Fifteen regional distributors of F&V were investigated. The city was chosen due to its few local suppliers and location (far from most producers). According to the literature, qualitative studies are recommended for research that aims to develop and offer detailed insights into organizational and individual processes [20,21].

Based on the literature review, a coding approach was chosen for the qualitative analysis of the text [22]. The procedure for coding the literature and the collected data was performed using ATLAS TI Software with the aim to unveil the causes of fruit wastage and its mitigators [23]. These causes and mitigators were coded as Storage, handling, and transport, and Other causes and mitigation.

The research questions of this study and the analysis of the causes and mitigating factors indicated in the literature guided the definition of the questions to be proposed. Since the interviewees were small entrepreneurs, the opening questions focused on causes and mitigants (existing or that could exist). Such focus aimed to improve the information collected during subsequent conversations. The questions used to guide the discussions on each code can be found in Table 1.

**Table 1.** Questions used to guide the discussions.

| Code | Question |
|---|---|
| Storage, handling, and transport | 1. How does storage contribute to fruit and vegetable waste?<br>2. How could the waste that occurs in storage be reduced?<br>3. How does handling contribute to fruit and vegetable waste?<br>4. How could the waste that occurs in handling be reduced?<br>5. How does transportation contribute to fruit and vegetable waste?<br>6. How could the waste that occurs in transport be reduced? |
| Other causes and mitigation | 7. Beyond storage, handling, and transport, what are the other causes of losses?<br>8. How could product waste related to these other causes be reduced? |

### 2.4. Data Collection

A pilot study embraced three distributors and aimed to validate the research instrument. After the pilot study, minor changes in the questionnaire were performed. The new questionnaire was then applied to the other interviewed professionals. The information collected with the semi-structured questionnaire constituted the primary dataset. The researchers also performed documental analysis and direct observations (for triangulation).

Table 2 presents the prominent aspects of the contributing distributors. The profiles of the individual respondents from F&V distributors are shown in Table 3.

**Table 2.** Profiles of distributors investigated.

| F&V Distributor | Founded in | Main Customers |
| --- | --- | --- |
| Distributor 1 | 1985 | Supermarkets |
| Distributor 2 | 2001 | Supermarkets and small restaurant chains |
| Distributor 3 | 1999 | Supermarkets and local F&V resellers |
| Distributor 4 | 2010 | Supermarkets |
| Distributor 5 | 2004 | Supermarkets and industrial restaurants |
| Distributor 6 | 2013 | Supermarkets |
| Distributor 7 | 2000 | Supermarkets |
| Distributor 8 | 1998 | Supermarkets |
| Distributor 9 | 2015 | Supermarkets |
| Distributor 10 | 2014 | Supermarkets |
| Distributor 11 | 1989 | Supermarkets |
| Distributor 12 | 2000 | Supermarkets and small restaurant chains |
| Distributor 13 | 1999 | Supermarkets and small restaurant chains |
| Distributor 14 | 2004 | Supermarkets |
| Distributor 15 | 1988 | Supermarkets |

**Table 3.** Profiles of respondents.

| Distributor | Position | Code | Experience | Interview |
| --- | --- | --- | --- | --- |
| Distributor 1 | General manager | D1 | 15 years | 67 min |
| Distributor 2 | General manager | D2 | 13 years | 53 min |
| Distributor 3 | Owner | D3 | 21 years | 45 min |
| Distributor 4 | Owner | D4 | 16 years | 67 min |
| Distributor 5 | General manager | D5 | 26 years | 60 min |
| Distributor 6 | Owner | D6 | 17 years | 44 min |
| Distributor 7 | General manager | D7 | 16 years | 40 min |
| Distributor 8 | General manager | D8 | 10 years | 80 min |
| Distributor 9 | General manager | D9 | 12 years | 60 min |
| Distributor 10 | General manager | D10 | 13 years | 63 min |
| Distributor 11 | Owner | D11 | 32 years | 56 min |
| Distributor 12 | Owner | D12 | 20 years | 45 min |
| Distributor 13 | General manager | D13 | 15 years | 88 min |
| Distributor 14 | Owner | D14 | 17 years | 50 min |
| Distributor 15 | Owner | D15 | 33 years | 44 min |

*2.5. Data Analysis*

After the interviews, the data were analyzed using qualitative content analysis. Notes taken during the direct observation of the facilities were also considered. The triangulation of all collected information data aimed to assure the reliability of the findings and the validation of the constructs [4,19,31]. Data were transcribed and coded using ATLAS TI Software, following the coding procedure [23]. The conclusions were cataloged into groups to identify the possible contributions of the findings to the academic literature.

The finding analysis allowed us to obtain a better understanding of the topic. Generalizability was addressed by selecting professionals who worked with distributors of similar

sizes. The validation of results and transferability were ensured by investigating managers or entrepreneurs that had developed actions to mitigate F&V waste. The attention paid to reliability was based on the benefits of these actions in reducing food waste, while confirmability was related to the individual analysis of each case. This analysis was carried out over three days, including all evidence collected. After analyzing each case individually, a cross-case analysis was performed using ATLAS TI software. This analysis aimed to identify similarities and differences between the interviewees, and the reasons for these similarities/differences. The results were coded to compare and contrast them with the elements extracted from the literature. The revised documents were then presented to the interviewees. Integrity was assured through anonymity and adherence to ethical standards.

## 3. Results

The causes identified seem to be driven by the investigated region's poor economic conditions (a common problem in emerging markets). Such causes appear to make investments by farmers and distributors difficult, thus helping to increase F&V wastage. In addition to the problems of the regional scenario, this study also identified sectoral gaps that, once filled, could help reduce F&V waste. The problems and gaps identified were codified in farmer planting exceeding the demand, transportation and handling, storage and packaging, and sales below expectations. Details are provided below.

### 3.1. Farmer Planting Exceeding the Demand

The most significant waste cause verifies when farmers plant greater quantities than retailer sales capacity (thus generating overproduction). Over-planting seems to be caused by the lack of information on demand tendencies or what other farmers plan to grow. Without such information, farmers must guess the demand or the quantities that other farmers produce. The documents provided by the distributors allowed us to triangulate such findings. As ascertained, the amounts offered by farmers exceeded the market demand in one year, while they were below the order in the next year (e.g., for tomatoes or lettuce). As a result, prices fluctuated.

When over-planting occurs, farmers reduce their prices (to mitigate losses). The documental analysis confirmed the price variation over the years. Lower prices lead distributors to buy large quantities (expecting to expand the profit margin). However, frustration due to sales of overproduced products leverages waste generation. The observations and documental analysis performed by the researchers confirmed that some products experienced large waste (compared with another product harvested in the same season).

According to the distributors, well-planned planting could mitigate over-planting and increase farmer and distributor revenue. However, most farmers do not have the necessary information to perform such planning (e.g., the demand tendencies or the amounts that other farmers plan to grow). Worse than that, in the distributors' opinion, knowing these data in advance would be useless to most farmers. According to the interviewed professionals, most farmers do not know how to use such information. In the distributors' opinion, farmers are experts in planting and harvesting exclusively.

When asked for a possible solution, some distributors suggested that sectoral entities (cooperatives or governmental agencies) should better support farmers. These entities could collect information on other farmers' plans or distributor demand and analyze it. Entities should also instruct farmers on how to use the results of such analyses (e.g., to define how much to produce). However, the distributors did not know of any entity that provided such services. The documental analysis and observations performed by the researchers confirmed the absence of such services in the area investigated; the lack of an entity that provides reliable data results in more significant planting quantities than the trade demands.

Success in the actions that an entity could develop requires attention to be paid to all actors in the chain. As stated by Distributor 4, farmers and distributors have gains and losses with waste. This is the case of the cost–benefit of planting more lettuce than required

(farmers) or buying items that exceed the demand (distributors). A better understanding of such gains and losses could induce farmers and distributors to change their practices.

### 3.2. Transportation and Handling

Problems in transportation (from the farm to the distributor and from the distributor to the retailer) or inadequate handling in warehouses can contaminate F&V. Problems in sanitizing, transportation, or handling resources also contribute to the accelerated maturation and senescence of F&V. These problems may accelerate deterioration, according to our observations.

According to the interviewees, the lack of attention paid to handling or transportation may damage products. The researchers' direct observations confirmed this finding (triangulation). These observations also indicated that only a few distributors handled F&V with care, used pallets to avoid contact with floor moisture, or performed reasonable temperature controlling. Transportation to the retailer constitutes another problem. During this ride, the distributor usually mixes, in the same truck, items that require different temperatures for conservation. This practice may accelerate F&V deterioration. The unfavorable socioeconomic factors in some parts of an emerging country may lead entrepreneurs not to invest in better transportation and handling. Such a stance favors the deterioration process, especially for tomatoes, bananas, papayas, and onions. In addition, mishandling while loading and unloading at distributor facilities generates friction or exposes F&V to moisture, thus compromising the integrity of F&V and their durability.

The findings indicate a few alternatives to mitigate the problems in the transportation and handling of F&V. According to the interviewed professionals, the whole food chain lacks adequate training to avoid waste. They believe that new procedures or training could prevent friction during transportation or handling and reduce exposure to moisture or temperature fluctuations. Due to the high turnover or low employee qualifications, (re)training must be periodic.

### 3.3. Storage and Packaging

Cheap or inadequate packaging may increase F&V damage. This is the case with fiber bags (for zucchini, potatoes, onions, chayotes, and cucumbers), wooden boxes (for pineapples), or plastic containers (for tomatoes and bananas). Storage problems can also be related to the facility size or conditions. The direct observations performed by the researchers showed that many distributors stored oranges, watermelons, and pumpkins in contact with the floor (which reduces their shelf life). The observations also indicated that many distributor facilities presented small sizes, poor conservation (floor, walls, and roof), or poorly lit and poorly ventilated environments.

As indicated in the Transportation and Handling subsection, the interviewed professionals did not suggest alternatives to mitigate the problems in packaging or storage.

### 3.4. Sales below Expectations

The information collected during interviews and the direct observations performed by the researchers indicated that even distributors with good practices and storage equipment may face F&V deterioration inside their facilities. Essentially, deterioration is a natural process for F&V. So, distributors can try to delay it but cannot stop it. No F&V last forever. In such a context, the findings indicate that distributors must reduce the impact of inevitable F&V deterioration.

The findings also indicate that last-mile distributors mitigate the negative impacts of deterioration. As ascertained, these distributors establish partnerships with retailers to sell excess inventories before their decline. The strategies to increase such sales include regulating prices according to the validity period. When these sales are not closed, small distributors donate or sell products at huge discounts to popular restaurants (which is coordinated by the municipal executive branch) or food producer cooperatives and

other derivatives. The direct observations and documental analysis performed during the triangulation confirmed such practice.

Table 4 presents the summary of findings.

**Table 4.** Causes of F&V waste and what we should know to mitigate it.

| Code | Causes | What We Should Know |
|---|---|---|
| Farmer planting exceeding the demand | Farmers plant higher quantities than the retailer sales capacity due to the lack of information on demand tendency or what other farmers plan to grow. | How can sectoral actors provide forecasts and support services to small farmers? |
| | | How can sectoral actors help farmers and distributors to use demand forecasts (when defining how much to plant or buy)? |
| | Exceeding the market demand generates gains and losses for farmers and distributors. | How can we change farmer and distributor practices based on the demonstration of financial gains and losses related to waste generation? |
| Transportation and handling | Problems in transportation, inadequate handling, or poor sanitizing or temperature control. | How can we induce entrepreneurs to invest in better resources or practices? |
| Storage and packaging | Inadequate packaging or storage. | How can we induce entrepreneurs to invest in better warehouses or packaging? |
| Sales below expectations | Even distributors with good practices face F&V deterioration inside their facilities. | How can we increase the alternatives to selling excess inventories? |

## 4. Discussion

### 4.1. Farmer Planting Exceeding Demand

This study contributes to the literature by indicating that sectoral entities can help to align better quantities planted by farmers and retailer demand [1,17]. To do so, greater cooperation from all chain actors may be required [17,39,40,45]. The position occupied by last-mile distributors in the chain allows them to cooperate in this planning improvement [1].

However, the literature is almost silent on how sectoral actors could support farmers and distributors. So, this study also contributes to the literature by suggesting that future studies should develop methods to reliably plan the future demand of the retailers served by a distributor. To generate better results in practice, we also need to know how to disseminate a plan throughout the chain and help farmers and distributors use demand forecasts. The analysis and dissemination of good information constitutes a gap not yet filled in the literature [23,43,44].

Over-planting reduction must focus on the farmer–distributor relationship. As ascertained, most planters are small or medium entrepreneurs who do not have the conditions to use the planning that the entities could provide. In addition, a farmer may serve different distributors. This study contributes to the literature by suggesting that further studies should investigate how farmers can plan their production in a scenario with several distributors, other alternatives of items to be planted, and profits related to such options (throughout the season). Developing a simulation tool that guides farmers when making such decisions could be valuable. By reducing losses due to overproduction, farmers and distributors could have more resources to address the other causes of F&V waste generation [22,24–26].

The success of such actions must consider the gains and losses that a farmer or distributors may have by reducing (or not) F&V generation. As ascertained, F&V waste may be perceived as a better alternative to generating profits in the short term. This study contributes to the academic literature by suggesting that not only distributors—as suggested in the literature [5]—but also farmers need to improve their planning.

### 4.2. Transportation and Handling

This study shows that unfavorable socioeconomic factors may hinder investments in better solutions or training (mainly in emerging markets). This contribution seems to justify why the actors in the chain do not improve their practices or resources [22,24,26].

Since the literature is almost silent on dealing with such issues, this study contributes to existing research by suggesting that further investigations should quantify the real impact of better resources on small entrepreneurs' profitability. By better understanding such numbers, other researchers could develop models that guide farmers, distributors, and retailers when investing in reducing F&V waste. The future use of such a model throughout the chain could help to improve infrastructure or practices. These demands are presented in the literature [4,15,28,38].

### 4.3. Storage and Packaging

According to the literature, storage practices and inadequate storage equipment or packaging seem to be very common worldwide [4,22]. Once again, the findings and the literature do not indicate alternatives to mitigate such issues at the level of small last-mile distributors. In addition, the literature is silent regarding the evaluation of the cost–benefit of better packaging or storage equipment.

This study contributes to the literature by suggesting that further studies should investigate the cost–benefit of more appropriate packaging or storage equipment. This information could help small distributors to better evaluate investing in such items [4,15,28,38].

### 4.4. Sales below Expectations

Despite all existing technology, F&V deterioration persists. This is a natural process that must be managed.

The findings indicate that distributors try to mitigate the negative impacts of this deterioration by selling items. A similar practice is used in supermarkets [1,46]. Trading before complete decline seems to be more sustainable than discarding non-perfect F&V [23,43,44]. However, such a possibility should be evaluated. This study contributes to academic research by indicating that further studies should analyze the environmental cost–benefit related to selling such items. Based on this information, governments could invest in sectoral entities or develop public policies that improve sustainability in the chain [1,17].

Success in selling items requires a well-designed sales and price strategy. However, the literature does not present a model to define win–win prices related to F&V waste mitigation (from farmers to retailers). Such a model could increase profits throughout the chain. Higher yields may increase the number of commercialization partners (thus leveraging sales and avoiding waste generation). So, this study contributes to the literature by suggesting that further studies should develop an evaluation model that uses historical data. Other studies could investigate how to prospect and sell non-perfect items stored at all points of the F&V chain [1,17].

## 5. Conclusions

This research aimed to identify the gaps that hinder F&V waste reduction. Special attention was paid to the last miles of said chain (due to the high environmental, economic, and social impacts of the losses at that point of the chain). The leading causes of waste were classified as farmer planting exceeding the demand, transportation/handling, warehouses/packaging, and sales below expectations. The findings contribute to existing research by showing a set of gaps related to each group of causes:

- To reduce the impact of farmer over-planting, we should know the following: How can sectoral actors provide forecasts and support services to small farmers? How can sectoral actors help farmers and distributors to use demand forecasts (when defining how much to plant or buy)? How can we change farmer and distributor practices based on demonstrating financial gains and losses related to waste reduction?

- To improve transportation/handling, we should know the following: How can we induce entrepreneurs to invest in better resources or practices?
- To improve warehouses/packaging, we should know the following: How can we induce entrepreneurs to invest in better warehouses or packaging?
- To increase sales below expectations, we should know the following: How can we improve the alternatives to selling excess inventories?

**Author Contributions:** Conceptualization, F.C.V.S., G.M.P., L.R.T., M.D.S. and G.S.M.; methodology, M.B. and F.C.V.S.; software, M.D.S., L.R.T. and G.M.P.; formal analysis, F.C.V.S., L.R.T., M.B. and G.S.M.; investigation, F.C.V.S. and G.M.P.; writing—original draft preparation, G.S.M. and M.B.; writing—review and editing, F.C.V.S., L.R.T. and G.M.P.; visualization, M.D.S.; supervision, G.M.P.; project administration, F.C.V.S. All authors have read and agreed to the published version of the manuscript.

**Funding:** This research received no external funding.

**Institutional Review Board Statement:** Not applicable.

**Informed Consent Statement:** Not applicable.

**Data Availability Statement:** Not applicable.

**Conflicts of Interest:** The authors declare no conflict of interest.

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
