# Peer review of "Food Waste in Distribution: Causes and Gaps to Be Filled"

_sustainability, doi:10.3390/su15043598_

Round 1

Reviewer 1 Report

In this manuscript, through literature review and questionnaire analysis the authors tried to find out ways to reduce the huge waste in fruit and vegetable distribution, and gave several suggestions. This research is meaningful. However, the survey section lacks detailed data. I suggest the authors provide the detailed data and data analysis of the interviews for better understanding the results and discussion.

Author Response

Dear Reviewer,

Thank you very much for your careful reading of our manuscript. You helped us to identify fundamental problems (which we overlooked during the manuscript preparation).

After analyzing your comments, we rewrote the abstract and sections 1, 2, 3, 4, and 5. 

We have paid particular attention to presenting our findings, mainly the ones that may contribute to the literature. The indication of these contributions has been improved in the discussion section.

Once again, thanks for your comments.

Reviewer 2 Report

Dear authors,

I revised your manscript. The outputs obtained from interviews are not providing new information in food waste reduction studies I also wanted to see the statistical analysisi resuÅŸts, including validations of the questions.  From this perspective the study seems having an average sientific knowledge and interest. 

Author Response

(The authors gave the same response as above.)

Reviewer 3 Report

The article was trying to investigate the food waste problem in the channel. It’s an important and interesting topic and the researchers conducted a qualitative research to examine the causes and potential solutions. I believe, though, that quite a bit still needs to be done to clarify its contribution and better explain the method. Below are my concerns/suggestions for improvement:

1.     My biggest concern is that it’s unclear what’s contribution this article makes to the existing literature. The authors provided a literature review of food supply chain and food waste causes but did not point out what was the gap that the research is trying to fill. Thus it’s not clear that how the current research differentiate from the existing research.

2.     Extending my comment #1, I would also suggest the authors to provide further explanation of how the interview questions were selected. Were they based on the existing research, or it’s relevant to what the research gap that the authors are trying to investigate?

3.     The RQ stated that “RQ: How to mitigate the fruit and vegetable waste in the chain's last miles?” but the interview questions are in fact asking about the causes. Therefore, it’s again a mystery to the reader what exactly is the research trying to investigate. Are you looking for the causes or the solutions? If the answer is solutions, then I would have to doubt whether the current research method serve the purpose.  It seems that the authors proposed solutions based on the findings of the causes.

4.     The authors mentioned that triangulation was “considered”, but did not provide enough details about how they conducted the triangulation.

5.     The authors should also at least provide a table of the analytical results of the contents to support your finding sections.

Minor things:

1.     Please briefly introduce the method in the abstract

2.     The title is double-negative, which is hard to understand for readers.

Author Response

(The authors gave the same response as above.)

Round 2

Reviewer 3 Report

The manuscript is substantially improved. They have made a good effort to address the issues in the last round. I will sign off on this one. 

Good job overall!